# Unmasking AlphaFold to integrate experiments and predictions in multimeric complexes

Claudio Mirabello [1] ✉, Björn Wallner [2], Björn Nystedt [3], Stavros Azinas [4] & Marta Carroni [4]

Since the release of AlphaFold, researchers have actively refined its predictions and attempted to integrate it into existing pipelines for determining protein structures. These efforts have introduced a number of functionalities and optimisations at the latest Critical Assessment of protein Structure Prediction edition (CASP15), resulting in a marked improvement in the prediction of multimeric protein structures. However, AlphaFold's capability of predicting large protein complexes is still limited and integrating experimental data in the prediction pipeline is not straightforward. In this study, we introduce AF_unmasked to overcome these limitations. Our results demonstrate that AF_unmasked can integrate experimental information to build larger or hard to predict protein assemblies with high confidence. The resulting predictions can help interpret and augment experimental data. This approach generates high quality (DockQ score > 0.8) structures even when little to no evolutionary information is available and imperfect experimental structures are used as a starting point. AF_unmasked is developed and optimised to fill incomplete experimental structures (structural inpainting), which may provide insights into protein dynamics. In summary, AF_unmasked provides an easy-to-use method that efficiently integrates experiments to predict large protein complexes more confidently.

Since the release of AlphaFold (v2)[1] in 2020, part of the computational structural biology community has worked to improve AlphaFold and to expand its functionalities, also in ways its creators had not initially envisioned. This is a challenging avenue of research, as it involves manipulating a deep neural network in ways that may yield unpredictable results. Interpretation of neural networks is also notoriously hard.

The authors of a recent study have theorised that the neural network performs a sampling technique on a learned energy landscape[2]. According to this theory, the first block of layers (the

Evoformer module) identifies a neighborhood within this landscape that closely approximates a reasonable minimum. The second block (the structural module), known for its ability to accurately predict the quality of predictions[1,2], performs an energy minimisation within the identified neighborhood to generate atomic structures. More recently, targeting the energy minimisation in this second step has been a main strategy to improve AlphaFold. The main focus at the latest edition of the Critical Assessment of protein Structure Prediction (CASP15) was to assess the progress in the assembly category since AlphaFold-Multimer[3] had been released. The best ranking groups applied mainly

[1]Dept of Physics, Chemistry and Biology, National Bioinformatics Infrastructure Sweden, Science for Life Laboratory, Linköping University, 581 83 Linköping, Sweden. [2]Dept of Physics, Chemistry and Biology, Linköping University, 581 83 Linköping, Sweden. [3]Dept of Cell and Molecular Biology, National Bioinformatics Infrastructure Sweden, Science for Life Laboratory, Uppsala University, Husargatan 3, SE-752 37 Uppsala, Sweden. [4]Dept of Biochemistry and Biophysics, Science for Life Laboratory, Stockholm University, Stockholm, Sweden. ✉e-mail: claudio.mirabello@scilifelab.se

two strategies: either they (i) introduced stochastic noise into the neural network while generating thousands of models for each target[4], or (ii) created and selected better/deeper Multiple Sequence Alignments (MSAs) as input to the Evoformer module[5]. If the prediction task is indeed one of energy minimisation, the first approach would help sampling AlphaFold's energy function more efficiently and extensively. The second approach could be interpreted as an attempt to start the sampling procedure in a lower-energy neighborhood. Both approaches aim to maximise AlphaFold's predicted quality score in order to sift out the best models. On the other hand, there seemed to be few gains made optimising the template inputs, containing structural information (in the form of atom coordinates) about homologous proteins structures from the Protein Data Bank (PDB), a source of information that is complementary to the MSAs. Some groups at CASP15 have gone as far as turning templates inputs off altogether[5]. Regardless of the sampling strategy, predicting large multimeric structures remains both challenging, as the probability of predicting incorrect interfaces increases with the number of units in the complex. It is also prohibitive in terms of computational costs, and current state-of-the-art predictors require many iterations[6] to assemble large multimeric complexes.

The wider structural biology community, outside of that of CASP, seems united in regarding AlphaFold as a useful tool that aids structure determination and density interpretation, but one that cannot yet replace experiments[7,8]. Indeed, a lot of work is underway to develop better integrative tools. For example, Phenix integrates AlphaFold within a more classical Molecular Replacement approach involving the trimming, breaking, and assembly of predicted monomers for refinement against experimental maps obtained by X-ray crystallography[9].

It has also been shown that it is possible to integrate cryogenic electron microscopy (cryo-EM) and X-ray crystallography experiments with predictions by iteratively refining AlphaFold models against experimental data and inputting the refined models in AlphaFold as structural templates[10,11]. This approach shows that templates can be used as a vector to inject experimental information into a prediction pipeline. Still, AlphaFold was initially engineered for monomeric predictions and even though several multimer versions have been released, templates in AlphaFold remain monomeric in the sense that any potential information about interactions in quaternary templates is not used (see Methods). Templates will help build each monomeric subunit in a complex, but not the multimeric assembly of the subunits.

Other integrative approaches propose retraining versions of AlphaFold that are better at taking experiments into account: versions of OpenFold[12] and Uni-Fold[13] (retrainable implementations of AlphaFold) accept extra input data derived from crosslinking experiments to guide monomeric and multimeric predictions[14]. Still, retraining a large neural network like AlphaFold requires a lot of resources, and results show that these predictors are not always as good as those that participated at CASP15[14].

In this work, we introduce AF_unmasked, a version of AlphaFold designed to leverage information from templates containing quaternary structures to its full extent without the need for retraining. The extra information derived from quaternary templates greatly reduces the complexity of the prediction problem, allowing for faster prediction times and more accurate quaternary models, in particular for large protein complexes.

Changes made to AlphaFold are depicted in Fig. 1. The changes involve non-parametric layers in the neural network, thus re-training is not needed. We demonstrate that this approach effectively biases the network to produce high-quality structures of complexes, up to ~10 thousand residues in size, even in the absence of any evolutionary information.

Additional features include the possibility to use quaternary templates for improved integrative modelling, i.e.to resolve clashes often found in models assembled manually based on experimental data. Furthermore, we explore the possibility to perform *structural inpainting* within AlphaFold to integrate evolutionary restraints inferred from the MSA to fill in missing areas in the structure. Based on AlphaFold's learned energy potential, our implementation may even allow to estimate the effect of mutations or conformational changes in these complexes.

## Results

We test AF_unmasked on a series of cases derived from the PDB, from a dataset made of challenging multimeric targets from CASP15 as well as cryo-EM datasets of large protein complexes. First, we show how the multimeric template information is particularly useful in cases where the standard version of AlphaFold is unable to build the true complex. Furthermore, we show that even when imperfect templates are used, e.g. multimeric templates with clashing interfaces or missing parts, AF_unmasked improves on these inputs by remodelling parts or filling in the gaps by structural inpainting.

### Proof of concept

The first question is whether AlphaFold is at all capable of using cross-chain information derived from multimeric templates to build protein complexes. After all, the neural network was not trained to take distances across chains in account when building assemblies. Therefore we perform a number of proof-of-concept tests on a common benchmark, i.e. the PDB benchmark set (see Methods).

This benchmark set is made of heterodimeric structures that have not been used in the training of the latest version of Alphafold-Multimer (v2.3). First, we set a baseline by running the standard version of Alphafold-Multimer with its default template strategy and all parameters set to their default, thus producing 25 predictions per target.

We then test twice AF_unmasked with different sets of templates. In this test, we use as templates in AF_unmasked the same deposited structures that we wish to predict (i.e. ideal templates). We perform the same prediction task twice, once with the default template strategy (Masked), once with the new template strategy (Unmasked). This simple test allows to assess whether cross-chain distance constraints from ideal templates can inform the prediction task. In order to assess the impact of including or excluding MSA information during the prediction task, we also run both combinations either while using the complete MSA or with MSAs that were clipped to include only the target sequences (i.e. deleting all evolutionary inputs).

We score all predictions against the natives with DockQ[15]. In Fig. S1, we show the distributions of DockQ scores for all 251 targets in each test. For each target and test, we only score the prediction with highest ranking confidence predicted by AlphaFold. The Masked predictions are generally better than the Standard predictions both with or without evolutionary inputs (MSA). This should not surprise, as inputting ideal (monomeric) templates should help AlphaFold in finding the right arrangement of units. Interestingly though, disabling evolutionary inputs will make predictions worse as the cross-chain evolutionary coupling information is lost.

Conversely, enabling our proposed template mechanism improves predictions, making them almost perfect when no evolutionary information is used. This demonstrates that cross-chain distance constraints from templates can inform the prediction task, and should be used whenever possible.

When comparing Unmasked predictions done with/without evolutionary inputs, we notice an interesting pattern in the resulting scores, as seen in Fig. S2: while the predicted ranking confidence tends to be lower when the evolutionary inputs are missing, whenever the confidence increases the corresponding predictions are also better in quality. The fact that the confidence is lower in most cases is a hint that AlphaFold is not blindly trusting the templates, i.e. the predicted quality scores are not biased by the template inputs. On the other hand, those cases where the MSA input might be noisy, possibly due to

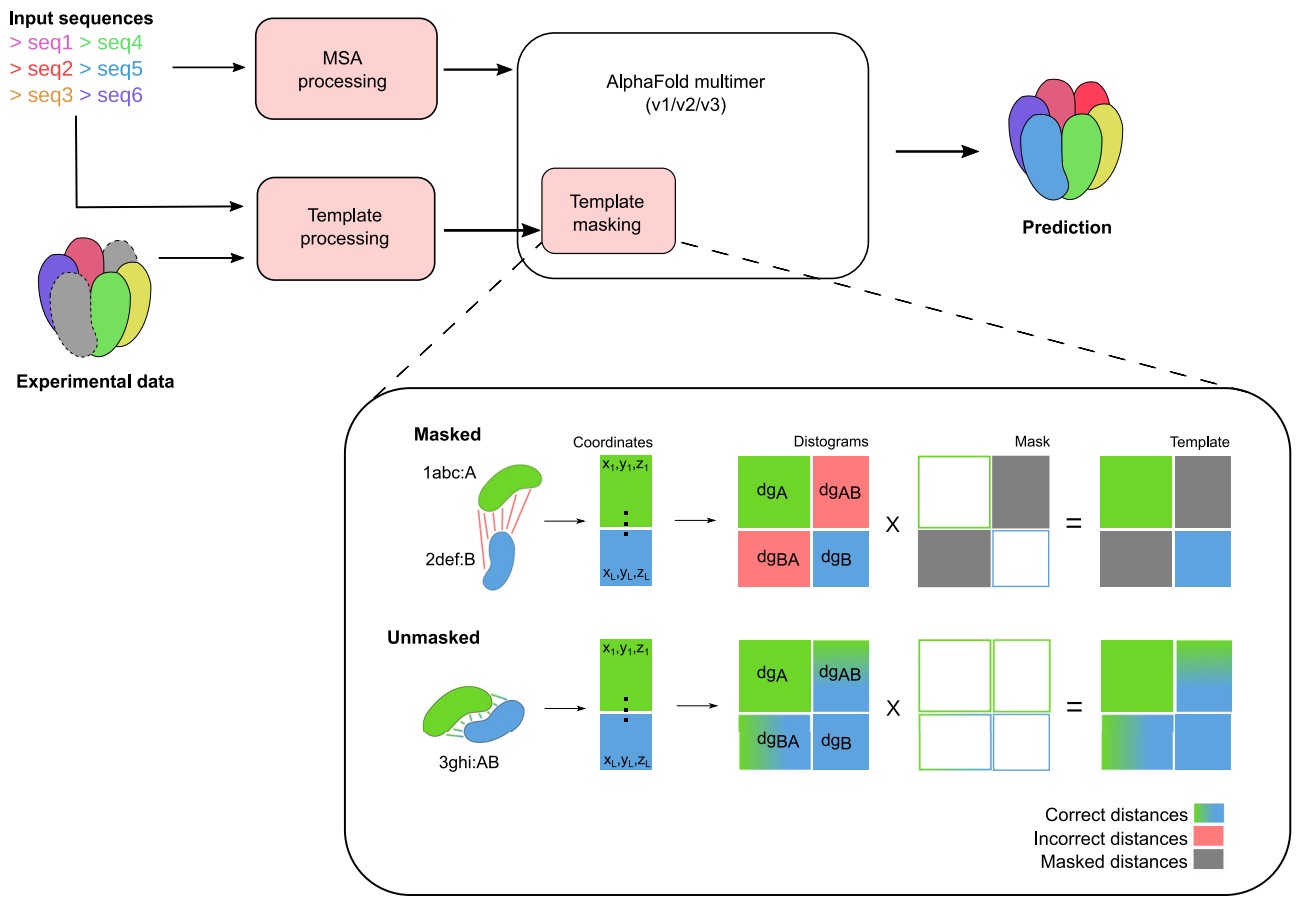

**Fig. 1 | Flowchart of AF_unmasked.** Structural data coming from an experiment is converted to the correct format (mmCIF), then the template is aligned, either by sequence against the target sequences or structurally against a set of target monomeric structures. This ensures that the template coordinates are applied to the right target amino acids, even when templates are remote homologs. At this stage, MSAs may be clipped to reduce their size and increase the influence of templates on the output. The templates are inputted in AlphaFold along with the evolutionary inputs (MSAs). In the "Template masking" close-up schema, we show changes made to the neural network in AlphaFold: by default, monomeric templates from unrelated structures have incorrect cross-chain distances, so the cross-chain masking, which is by default enforced in AlphaFold-Multimer (Template masking block, Masked track). Coordinates from different templates are merged in single Distograms ($dg$). Distograms contain information about distances within each monomer ($dg_A$, $dg_B$) as well as cross-chain distances ($dg_{AB}$, $dg_{BA}$). Cross-chain distances are filtered with a masking layer so that, in the final distogram, they are ignored by the neural network. When multimeric templates from experimental datasets are used, the distances across chains are correct and informative, so we disable the template masking (Unmasked track). The neural network then is informed by distances across chains as well as within chains.

lack of cross-chain evolutionary signal, the predicted quality score jumps up as soon as this input is eliminated, resulting in higher quality predictions as AlphaFold relies more on the templates. This means that the change in the neural network does not preclude the possibility to use AlphaFold's predicted quality scores to sift out good predictions, even in comparison to those obtained from the standard implementation of AlphaFold-Multimer.

## Homology modelling

Since ideal templates are often unavailable, we also assess AF_unmasked in the case where homologous templates that are at least somewhat informative are used instead. In this test we use the Homologous PDB set (see Methods) to produce predictions for 28 challenging targets where AlphaFold-Multimer cannot produce a correct top-ranked prediction. We follow the same testing protocol as described above, predicting from homologous templates with or without MSA information.

In Fig. 2 we compare predictions on these hard targets from homologous templates (Unmasked-Homologs) to those from the previous test. These predictions show a marked improvement over the Standard and Masked predictions from the previous test, underlining the usefulness of providing AlphaFold with cross-chain information

from homologous templates. Moreover, predictions from homologous templates are only slightly worse than those made from ideal templates (Unmasked, Fig. 2).

In Fig. S3 we compare the predictions generated with homologous templates against standard AlphaFold predictions and AF_unmasked predictions based on ideal templates. The figures show that there is no clear correlation between the quality of the predictions and the sequence identity between target and template sequences. This indicates that sequence similarity is not biasing the predictions, and that AF_unmasked is useful even when using remote homologs. Turning the evolutionary inputs off (no MSA predictions) on this dataset does not seem to have much of an impact on the quality of the predictions. We also test turning dropout on in this scenario, but given the limited number of predictions generated (only 25 per target) we see a small improvement on the overall quality that is not statistically significant.

## Using imperfect templates

Next, we test AF_unmasked on templates that are a coarse representation of a protein complex. This is a common scenario when performing molecular replacement or fitting densities in experiments, where users might generate predictions separately for unbound monomers and manually dock them according to the data. These are

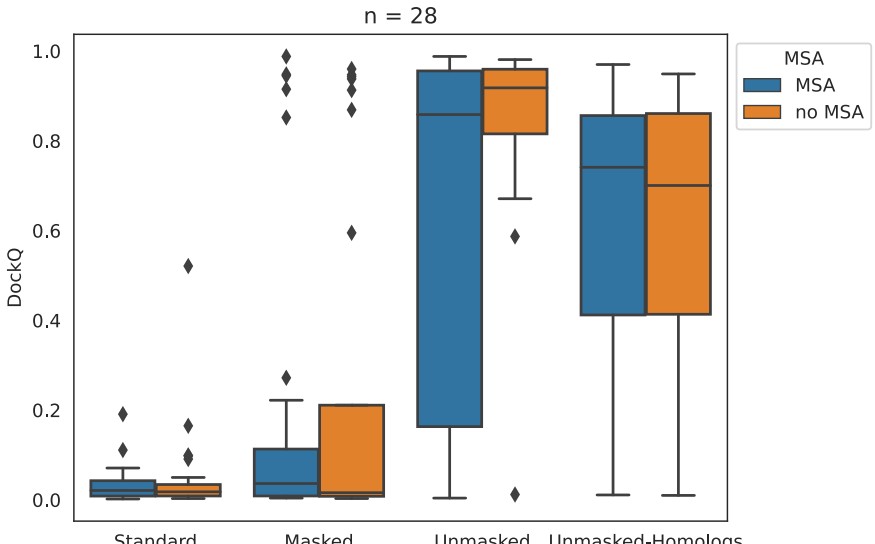

**Fig. 2 | Box plot comparison of various template strategies when predicting a subset of the PDB set of heterodimeric complexes.** Each box represents the inter-quartile range (IQR), with the median represented as a horizontal line. Whiskers extend to up to 1.5 × IQR beyond the box. Diamonds represent outlier samples. The subset in this test is made of heterodimers ($n = 28$) where good homologous templates could be found in the PDB and the predictions by AlphaFold-Multimer (Standard) are incorrect. We evaluate AF_unmasked on ideal, native templates without and with cross-chain restraints (Masked and Unmasked, respectively). Then we switch from ideal to homologous templates (Unmasked-Homologs). Only one the top-ranked prediction by ranking confidence, out of 25, is evaluated for each heterodimer. Though results are slightly worse than when providing an ideal template, the cross-chain information from homologous templates helps making better predictions than on Standard and Masked predictors.

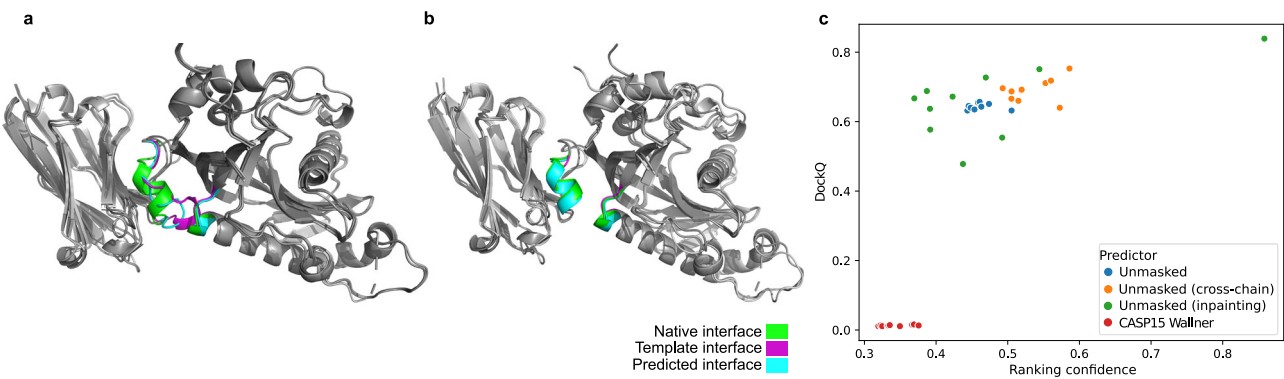

**Fig. 3 | CASP15 target H1142 is an antibody-antigen complex.** The template was obtained by superimposing unbound structures from CASP15 predictions onto the native to simulate an imperfect template. **a** In this case, some of the residues at the interface are clashing in the template. We test AF_unmasked either by feeding this imperfect template (**a**) or by deleting the clashing interfacial residues to let AF_unmasked inpaint them (**b**). Results show that (**c**) using both cross- and intra-chain restraints (Unmasked) from the imperfect template does not perform as well as using cross-chain restraints alone (Unmasked, cross-chain). The best overall strategy is to delete the clashes and perform inpainting (Unmasked, inpainting), which results in more extensive sampling of the space of conformations. Regardless of the strategy, the best model by ranking confidence was also the best model by DockQ.

rough models that might include, e.g. clashes at the interface or loops that are incorrectly modelled. We want to test whether AlphaFold can generate a correct structure from such imperfect templates. We perform this test with CASP15 target H1142, an antigen-nanobody complex with stoichiometry A1B1. This type of complex is a good test case, as AlphaFold is relatively weak in modelling antigen-antibody interactions[3], likely due to a lack of evolutionary coupling signal in the MSA[16].

We take AlphaFold predictions for each chain from incorrect models of the complex that were submitted by NBIS-AF2-multimer and position them in a roughly correct manner by superimposing each prediction to the corresponding unit in the experimental structure. The template obtained this way has DockQ score of 0.64 (medium quality) against the experimental structure. Since the chains were

extracted from incorrect predictions, even when they are positioned correctly with respect to each other, the interface is incorrect: the RMSD of interfacial residues (iRMSD) is 2.1 Å and 12 interfacial residues are clashing according to DockQ (Fig. 3a). We input this imperfect template into AF_unmasked while clipping the MSAs to a single sequence and predict 500 structures for each test. Results show that, when using both intra-chain and cross-chain restraints (Unmasked), the clash is fixed but the interface is not perfect. In a second test, we turn off intra-chain restraints while keeping only cross-chain restraints active. This should allow AlphaFold to rearrange each monomer wherever necessary while keeping the distances between the chains within the boundaries of the template, which results in a more diverse set of predictions (Fig. 3c) and a much-improved interface. Lastly, we generate a third set of predictions by letting AF_unmasked

automatically detect and delete clashing sets of residues in the template structures during the template preparation step (by setting the appropriate flag -inpaint_clashes). These deleted residues are regenerated (inpainted) during the prediction step. This seems the best approach, resulting in even more diverse predictions. Here, the prediction with highest ranking confidence is also the best overall model (Fig. 3b). Results for the whole set of 1500 predictions are shown in supplementary Fig. S4.

The correlation between AlphaFold's ranking confidence and the prediction quality measures, already noticeable when using the DockQ score in Fig. 3b, is almost perfect when evaluating the RMSD of the interfacial residues (iRMSD) as calculated in DockQ (Fig. S5). Since the change in the interface is rather subtle between the original clashing template and the desired configuration, observing the iRMSD better highlights how AlphaFold is able to recognise correct interfaces and rank them accordingly. Inpainting the interface allows AlphaFold to explore more configurations and find the best possible, both by ranking confidence and overall quality.

We also assess whether AF_unmasked can retrieve the correct conformation from imperfect templates obtained by perturbing the position of one chain with respect to the other. The perturbation is done by taking the native complex and running RosettaDock[17] 300 times with a dock perturbation flag so that the two monomers are randomly roto-translated from the initial position following a normal distribution centered at zero and with standard deviations of 5 Å for the translation and 11 degrees for the rotation ( −dock_pert 5 11 flag). This generates a set of templates of varying quality, depending on the magnitude of the random perturbation. We evaluate the initial quality of these artificially perturbed templates by DockQ score against the native conformation. We then run AF_unmasked 300 times by using a different perturbed template each time and score the best prediction, as ranked by AlphaFold's predicted ranking confidence, with DockQ. Since with H1142 evolutionary information is not useful to make a good prediction, we clip the MSAs in this case as well and let AF_unmasked rely on the perturbed template alone. This will give an idea of how close to the correct conformation the template needs to be to get a good prediction. In Fig. S6a we show the results from this test by comparing the initial DockQ score for a perturbed template and that of the highest confidence model generated from that template. Each point is a perturbed template, colored by its quality if it were scored according to the Critical Assessment of PRedicted Interactions (CAPRI) criteria[18]. According to such criteria, the perturbed templates have qualities ranging from Incorrect to High. Results show that AF_unmasked is always able to take a template of medium quality (initial DockQ score ≥ 0.49[19]) or better and use it to generate a high quality (DockQ score ≥ 0.8) prediction. Predictably, as the template quality degrades, so does the quality of the predictions. Still, AF_unmasked generates high quality predictions for 129 out of 171 templates of Acceptable quality (initial DockQ score ≥ 0.23) and for 10 out of 60 templates of Incorrect quality. As perturbed templates get farther away from the right solution (template DockQ score < 0.19), AF_unmasked fails to generate good predictions. In Fig. S6b we show the template with lowest initial quality (template DockQ score: 0.19) where AF_unmasked could still predict the correct conformation (prediction DockQ score: 0.87).

## Inpainting of very large structures

A known limitation of AlphaFold is its capability to generate models for large proteins, mostly due to computational limitations in terms of GPU memory. This is, of course, a significant limitation as many interesting protein complexes are large.

For example, CASP15 target H1111 is a 27-mer with stoichiometry A9B9C9 and 8460 amino acids in total. DeepMind, who modelled the complex post-CASP15 on a more efficient version (v2.3) of AlphaFold than was available to the public, could not perform the modelling of target H1111 in one go and assembled multiple structures with A3B3C3 stoichiometry instead by using a template (PDB ID: 7ALW) as guide[20]. Here, we show that it is possible to overcome this limitation with AF_unmasked while limiting the depth of the final MSA to a maximum of 512 total sequences.

We use again the deposited structure itself (PDB ID: 7QIJ) as template. This is a hard prediction task, as the first 362 residues of the largest subunits (C9) from the membrane-bound domain in the complex are not in the deposited structure, so AlphaFold has to inpaint this gap leveraging the evolutionary information coming from the MSA inputs. We generated 25 structures following this protocol (results shown in Fig. S7), which takes around 10h of GPU time per predicted structure (NVIDIA A100, 80GB RAM), and select the top three models by ranking confidence. As we can see in Fig. 4, the portion of the structure that is covered by the template stays the same across the three models, while the inpainted membrane region (in green) appears in a variety of conformations, from closed to open. This is, to the best of our knowledge, the largest structure ever generated in one shot using AlphaFold.

## Predicting the impact of mutations

AlphaFold is not trained to predict the effect of mutations on the folding of a protein, and it cannot predict the impact of single-point mutations on protein stability[21]. This might be due to the fact that few mutations on a target sequence result in virtually identical MSAs being used as input, which might mislead the neural network into inferring incorrect restraints.

For example, T1110o and T1109o are two closely related homodimeric CASP15 targets. They are, respectively, a wild-type and a mutant construct of Isocyanide hydratase, where the single-point mutation D183A causes a rearrangement of the C-terminus loops at the interface, as shown in Fig. 5a/b. We test whether AF_unmasked is capable of correctly switching between the two loop configurations by encouraging sampling around the region of interest. In order to do this, we use a structural template, obtained by looking for structural homologs in the PDB, where 20 residues in the loops in question are missing (PDB ID: 4K2H). The RMSD between the template and the native, excluding the loops, is 2.1 Å. The sequence identity between the target and template sequences is below 20%, so we align target and template structurally with TM-align. We clip the input MSAs to a single sequence, which means that AlphaFold should follow the template wherever possible, and attempt to model the loops ab initio since neither structural nor evolutionary inputs are given in that area of the structure.

Results show that for both T1110o (wildtype) and T1109o (mutant), AF_unmasked correctly arranges the loops in the model with highest ranking confidence (Fig. 5). In Fig. 5d, we compare DockQ scores for the top 10 T1109o models by ranking confidence against the models submitted by Wallner at CASP15. The top-ranked AF_unmasked model for the mutant is the best overall (DockQ: 0.804). We also test the default intra-chain constraint setting (i.e.: Masked) when using the same template and find that none of the top 5 models beat the new template strategy, while two correct models are in the top 10 (best DockQ: 0.803). Results for all predictions are shown in Fig. S8 and S9.

This suggests that including cross-template information puts AlphaFold closer to the correct solution, thus allowing for better sampling of the remaining space of configurations.

Since the changes in the structure are fairly subtle as the loops rearrange, we also show how AlphaFold's average predicted LDDT scores (pLDDT) in the inpainted loop regions alone are highest in the models with highest quality both for T1109o and T1110o (Fig. S10 and S11). This confirms that in cases where the best loop arrangement is unclear, or where mutations might cause local variations in the structure, the inpainting procedure produces better models, provided that the increase in quality is reflected by the predicted quality scores in the areas of interest.

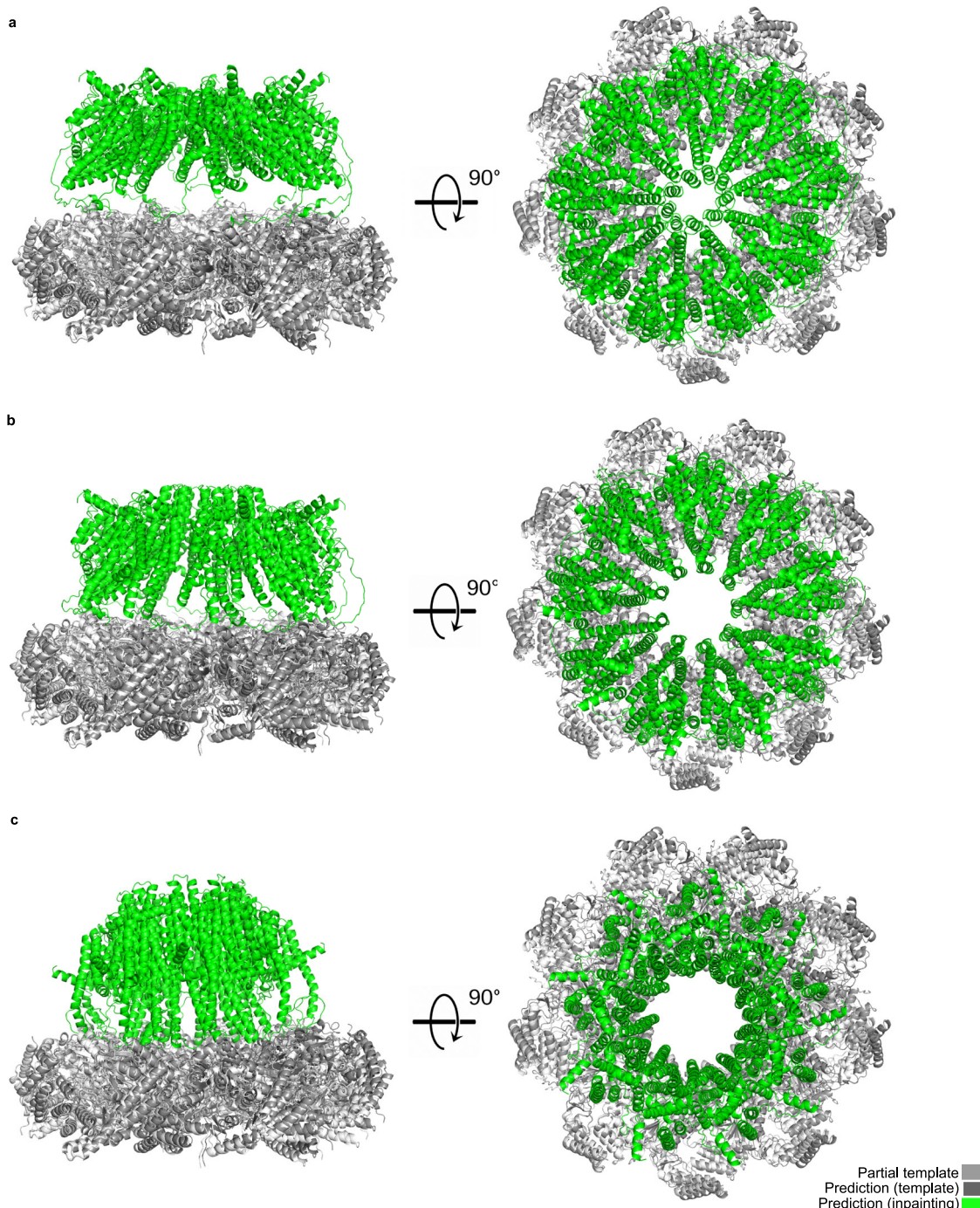

**Fig. 4 | CASP15 target H1111 is a very large complex (27 chains, 8460 amino acids) of a secretion export gate from *Yersinia enterocolitica*.** We use the CASP15 native structure (PDB ID: 7*QIJ*) as partial template (bottom ring) to guide the assembly and let AlphaFold inpaint the trans-membrane region. The top three models by ranking confidence are all near-identical to the template in the area covered by it, while the trans-membrane region show diverse and potentially biologically relevant conformations: closed (**a**), intermediate (**b**), open (**c**).

In Fig. 5c, we compare the models obtained for T1110o on the new template strategy against those submitted by Wallner in CASP15 and see that even in this case, the top selected model is also the one with the highest DockQ (Fig. 5a). Two of the top Wallner models have slightly higher DockQ.

## Cryo-EM test cases

In each of the biological systems we used, Rubisco, ClpB and Neuro-fibromin, we aimed at inpainting missing regions and identifying areas of possible conformational variability. We interpreted the results based on previous biochemical, structural and biophysical knowledge. Tens to hundreds of models were generated with AF_unmasked for each of the cases analysed, but we only display some representative examples for ease of description. The full set of models obtained, along with those shown in the figures and discussion, are available in the supplementary material.

**Rubisco.** Rubisco plays a crucial role in $CO_2$ fixation, making it responsible for the majority of organic carbon in the biosphere. Understanding the function and control of Rubisco remains a

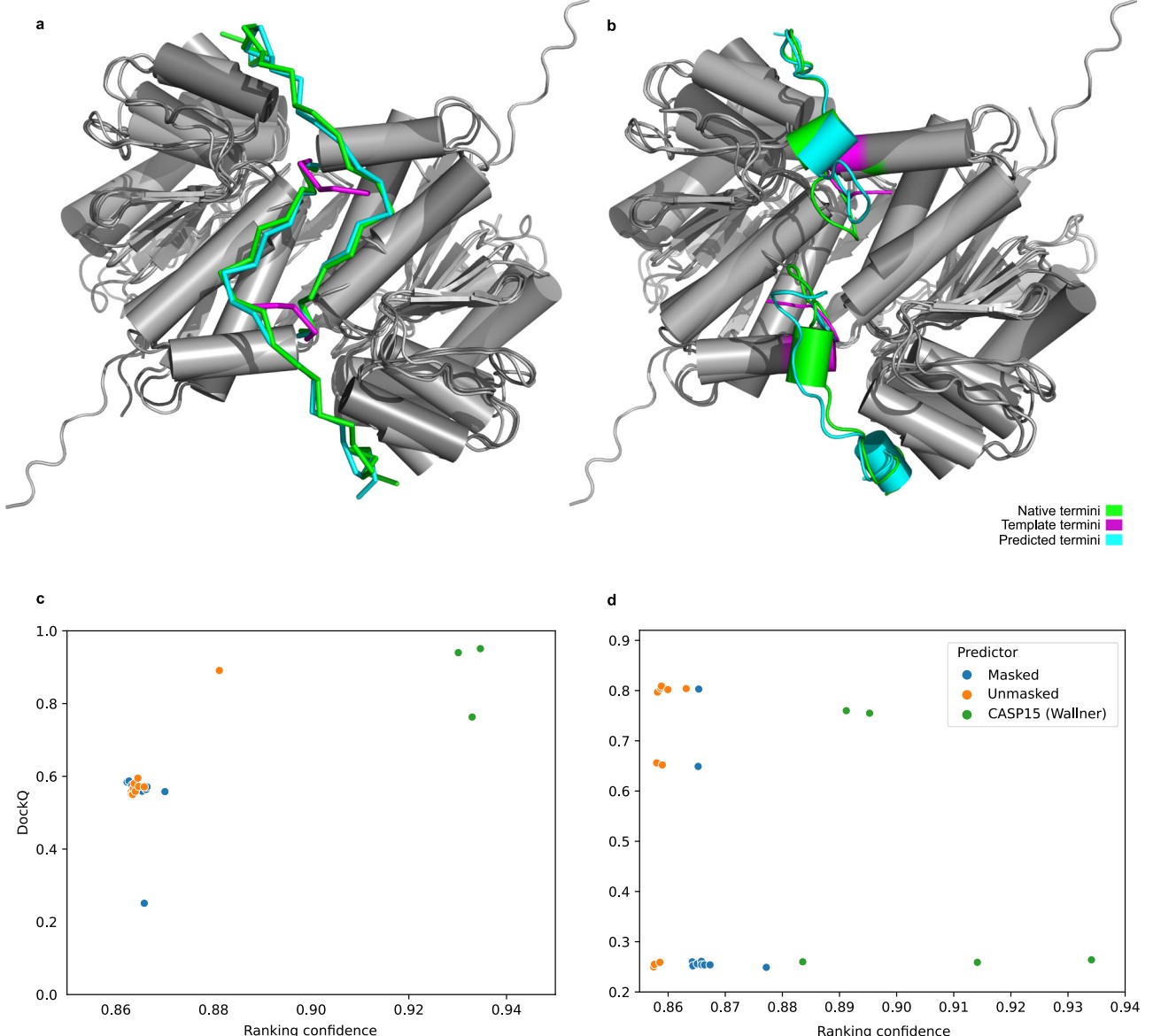

**Fig. 5 | CASP15 target T1110o is a homodimer of the isocyanide hydratase.**
**a** Target T1109o is a mutant of T1110o where a single-point mutation causes a rearrangement of the C-termini (**b**). The template was obtained by homology against the PDB, and among a set of candidates we selected a template where the C-termini loops were missing entirely (**a**, **b**). We utilise this template as is, the mapping between target and template amino acid sequences was performed by structural superposition between unbound models and the template with TM-align.

Using this incomplete template allows AF_unmasked to perform sampling of a number of different loop conformations through inpainting. The top-ranking structures by confidence score show the correct loop arrangement both in T1109o (**c**, Unmasked) and T1110o (**d**, Unmasked) for mutant and wildtype sequences, while the default template strategy (Masked) tends to assign to the mutant the same arrangement as in the wildtype.

significant area of research, with the aim of enhancing photosynthesis efficiency in agriculture and green biotechnology. The most prevalent form of Rubisco (Form I) comprises eight large and eight small subunits, and it exists in plants, algae, and other organisms. Although the active sites of Rubisco are situated in the large subunits, the expression of the small subunit regulates the size of the Rubisco pool in plants and can impact the overall catalytic efficiency of the Rubisco complex. For this reason, the small subunit is a potential target for bioengineering and biochemical studies have been performed to generate chimeras of large and small Rubisco subunits that could enhance Rubisco's performance[22].

We use AF_unmasked to predict a chimera of Rubisco composed of large subunits from *Arabidopsis thaliana*[23] and small subunits taken from another organism. We use a cryo-EM reconstruction obtained from own data (resolution: 2.06 Å) for this chimera molecule to assess

the quality of predictions from the standard version of AlphaFold-Multimer (v2.3) and from AF_unmasked. We are particularly interested in how well AlphaFold can predict the small subunits, as the inner loops of the subunits were challenging to reconstruct from the experimental density. The standard predictions do not agree with the experimental data in the area of interest, as the inner loops appear in a tighter conformation when compared to the experimental model density map (Fig. 6a, area circled in yellow). So we attempt a modelling step with AF_unmasked to improve this prediction. In this test, we provide AF_unmasked with the deposited structure from *Arabidopsis* (PDB ID: 5IU0) as template and let AF_unmasked transfer the homologous information from the template onto the chimeric sequence. We also delete a stretch of 20 amino acids from the inner loops in the PDB template in order to let AF_unmasked inpaint this region. Results (Fig. 6a, right) show that the best AF_unmasked inpainted model by

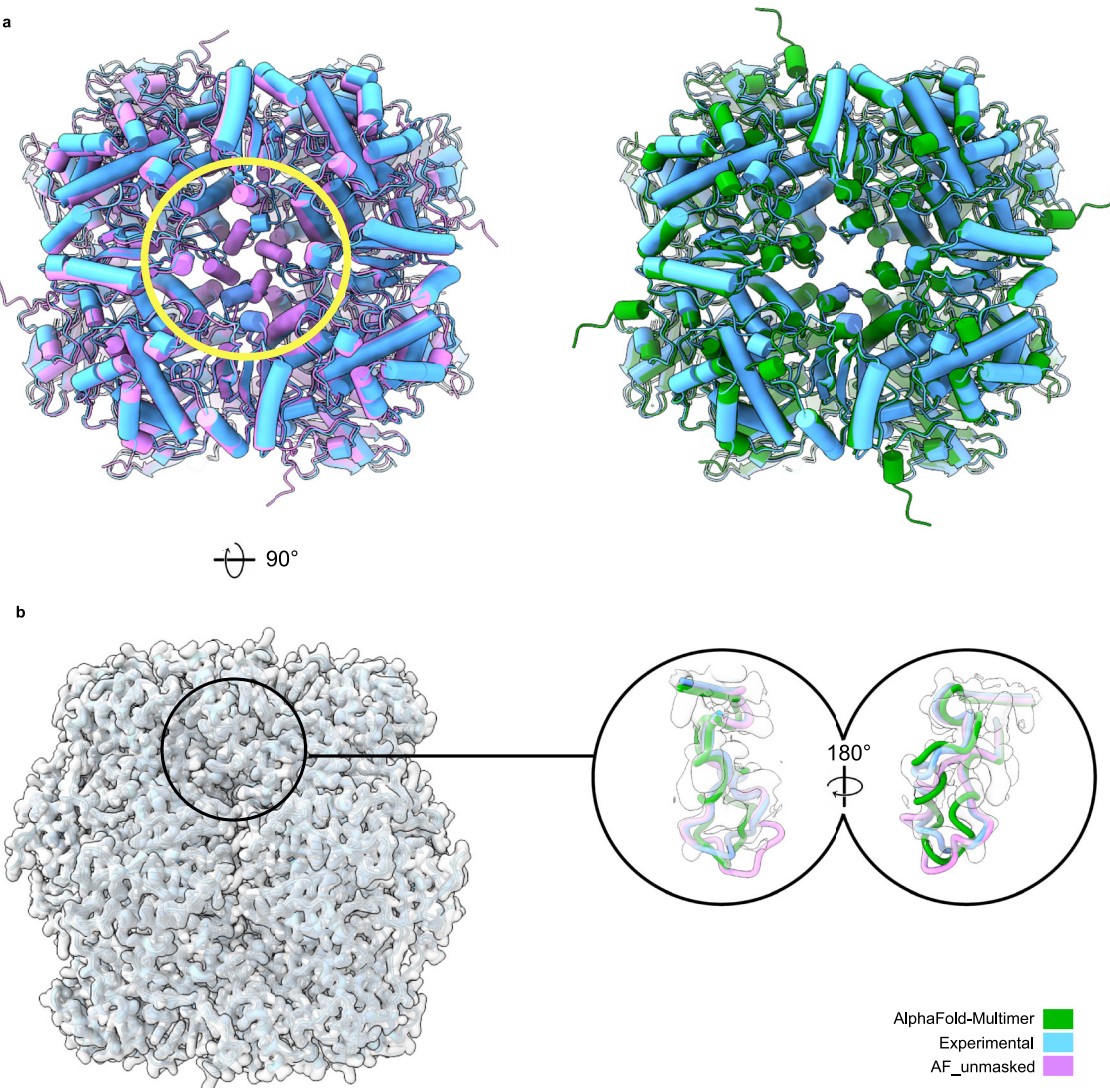

**Fig. 6 | Comparison of AF_unmasked and standard AlphaFold-Multimer predictions of chimeric rubisco protein.** Flexible loops in the smaller subunit at the center of the complex have been inpainted with AF_unmasked. **a** Global superposition of best standard AlphaFold-Multimer (v2.3) prediction by ranking confidence on the experimental cryo-EM structure (left) and comparison with the best AF_unmasked prediction (right). The circled area highlights how the inner loops are predicted in a tighter and symmetrical conformation compared to the experimental model. The AF_unmasked model, where the same inner loops were inpainted, shows better agreement with the experimental model. **b** Comparison of predictions against the density obtained from cryo-EM data after optimisation of the superposition between predicted loops and one loop from the deposited model. The circled area shows a cross-section of one of the inner loops of interest. The resulting inpainted loop fits better within the density and is a closer match to the final refined model when compared to the standard prediction.

ranking confidence is closer to the experimental structure in the area of interest when compared to the standard AlphaFold-Multimer predictions. The small subunits predicted by AF_unmasked fit better within the EM density, with a Q-score of 0.8 (comparable to that of the experimental structure: 0.81) which is higher than that of the standard AlphaFold-Multimer prediction (0.41) and are a closer match to the experimental structure in the inpainted loop region (Fig. 6b).

We also assess if the average pLDDT for the stretch of inpainted residues in the loop correlates with the quality of the predicted small unit. In Fig. S12 we show that inpainted loops with higher pLDDT will coincide with predictions that have lower RMSD when superimposed with the best matching chain from the experimental model.

**ClpB**. The bacterial chaperone ClpB is able to recover proteins from large aggregates and, together with the cognate DnaKJ system[24], to refold them into their active form. ClpB plays a pivotal role in protein homeostasis of bacterial cells (reviewed in[25,26]). ClpB works by using

ATP hydrolysis power to thread aggregates into a channel made upon oligomerisation of six identical copies[27]. The ClpB hexamer is therefore a very dynamic complex[28,29] that needs to recognise and then move to unfold the aggregated substrate.

A wealth of cryo-EM structural information about the ClpB hexamer is available[30–33], but large and highly-mobile domains are poorly defined. There are over 30 cryo-EM structures of the hexameric ClpB and eukaryotic homologue Hsp104, plus several X-ray crystal structures of ClpB/Hsp104 monomers, deposited in the Protein Data Bank. The ClpB crystal structure with PDB ID 1*QVR* shows two different localisations of the N-terminal domain relative to the rest of ClpB body and this is in good agreement with a number of biochemical studies showing that the N-terminal domain is involved in engagement of the misfolded substrate that will be then threaded through the ClpB hexameric channel[26]. In few cryo-EM maps, out of the six N-terminal domains, only two or three are visible at resolution lower than the rest of the ClpB body[33], thus indicating a high flexibility of this region. The

dynamic nature of the N-terminal domains has been shown also for systems analogous to ClpB, such as ClpX, ClpC, ClpA, 26S proteasomes, VCP among others.

We gave AF_unmasked a template derived from cryo-EM data (PDB ID: 5*OG*1) and performed inpainting of the missing regions. A total of 50 predictions were performed. Results show that, not only AF_unmasked could inpaint the missing N-terminal domains, but it also predicted them in multiple conformations within the same hexamer. This is different to AlphaFold-Multimer predictions, which are always highly symmetrical (Fig. S13a–b). Furthermore, the ranking confidence of AF_unmasked predictions correlates almost perfectly with the DockQ score of all interfaces in the hexamer (Fig. S14).

Another flexible ClpB region is the coiled-coil M-domain, known to be important in the regulation of ClpB and in its interaction with DnaKJ[34–36]. This is present in most ClpB structures, at least partially, and mutational studies[35,37] show that this domain can assume many different orientations and that such orientations are related to activation states of the ATPase. These orientations are reflected in the inpainted models, together with intermediate conformational steps, showing tilting of the long coiled-coil in ways that are plausible given the existing structural information, single molecule FRET-spectroscopy data[37,38] and coarse-grained Molecular Dynamics (MD) studies[35] (Fig. 7a–b).

To investigate the ability of predicting the interaction between ClpB and the commonly used substrate casein, we used a PDB template (PDB ID: 6*RN*3) where a stretch of amino acids from casein is engaged inside the ClpB pore. While most of 50 predictions generated this way failed to dock the casein inside ClpB (Fig. 7e), the top models by ranking confidence do show casein engaged inside the pore, and in one model, casein is in contact with one of the six N-terminal domains (Fig. 7c–d). The interaction between casein and N-terminal domain is predicted via hydrophobic patches (Fig. 7d), in good agreement with NMR data[39]. Predictions where the ClpB interacts with casein have higher pLDDT scores than those where ClpB is not engaged with casein (Fig. 7e). Once more, the ranking confidence correlates well with the DockQ score of all interfaces (Fig. S15).

**Neurofibromin.** Neurofibromin (NF1) is a downregulator of the oncogenic protein RAS and is ubiquitously expressed in the central neural system[40]. Neurofibromin plays therefore a very important role in tumor growth and its mutations cause the pleomorphic disease neurofibromatosis type 1[41] and are found in up to 10% of all cancers[42].

NF1 is an homodimer of around 600KDa and has a unique oligomeric arrangement made of a bi-lobate platform, composed of ARM and HEAT repeats[43–46]. The RAS-binding domain GRD and the membrane-binding domain Sec14-PH are anchored to this helical platform by long loops that allow large movements of these domains[43,45]. Cryo-EM structural studies[43,45,46] show two main conformations of the GRD and Sec14-PH domain that go from a so-called closed auto-inhibited conformation to an open conformation, which can bind RAS.

The standard version of AlphaFold-Multimer fails to find the right conformation of both monomeric and dimeric NF1 arrangements. In order to test whether AF_unmasked could reproduce the movements of GRD and Sec14-PH that are visible in the experimental data, we decided to remove these domains from the available deposited PDB structure (PDB ID: 7*PGU*) and perform inpainting on them. Given the considerable size of the inpainted domains, as well as the considerable scale of the movements involving these domains, we perform additional sampling and produce roughly 1000 models. Results show that AF_unmasked, using the bilobate helical part of the complex as template, was able to inpaint the missing GRD and Sec14-PH domains (Fig. 8a) as well as all the loops missing in the cryo-EM models (Fig. 8b). The inpainted domains are in good agreement with the cryo-EM and the crystal structures of the GRD and Sec14-PH domains (PDB ID: 6*OB*3,

1*NF*1, 3*PEG*, see also Supplementary Data). Results also shown good correlation between ranking confidence and DockQ scores (Fig. S16).

However, possibly due to the complexity of this inpainting task and the fact that the closed, auto-inhibited conformation of NF1 is stabilised by a coordinated *Zn* atom that cannot be taken in consideration here (Fig. 8b), none of the predicted models captures the state with one monomer in the closed conformation and one in the open, as seen in one of the deposited structures. On the other hand, the predicted conformations of GRD-Sec14-PH could be intermediate states of transition from the closed to the open NF1 conformations found experimentally in the cryo-EM studies. Indeed, the structures predicted this way can be used to aid in generating a morphing between the closed and open positions of GRD and Sec14-PH, which looks very different and structurally more likely (Supplementary Movies 1–2) than the simple morphing from closed to open[43]. These movements are only interpolations between predicted states that cannot be confirmed experimentally yet, but they represent solid hypotheses to be confirmed with further biochemical experiments and molecular dynamics simulations.

Interestingly, depending on the neural network model, different types of variability are picked. For example, we were also able to model different bending states of the helical platform (Fig. 8d) that are in good agreement with the 3D variability analysis performed in cryo-EM (Supplementary Movie 3).

## Discussion

AlphaFold has had a huge impact on the field of structural biology, but both computational and structural biologists are yet to maximise its potential, especially when it comes to modelling large protein complexes. Furthermore, integrating AlphaFold with other pipelines for the determination of protein structures is still an open question.

Approaches followed by computational biologists, providing the Evoformer module with better or more diverse sets of MSAs or performing more sampling, are promising avenues of research but only focus on the evolutionary aspect of the prediction problem while overlooking templates. But MSAs and templates are complementary sources of information: the first allows to derive evolution-driven restraints between pairs of amino acids, while the second is a direct observation of 3D structures.

If the prediction task in AlphaFold is like searching through a learned energy potential in order to find the bottom of the funnel, restraining the search through the use of experimental models will greatly reduce the complexity of the search.

The template mechanism in AlphaFold takes PDB structures and converts them in distograms, data structures containing information about distances between all atoms in a structure. The template distograms are used in the Neural Network models to provide an initial bias term to some of the attention mechanisms in the EvoFormer transformer model[1]. Providing a good template, then, actively biases AlphaFold in paying attention to specific pairs of interacting amino acids and allows the search to start from a close-to-optimum area of the landscape of conformations. This allows to find the bottom of an energy funnel just as well or better than a deep and informative MSA. This is especially true when it comes to protein complexes, in particular larger ones or those containing a large number of protein chains, where the evolutionary signal linking different protein chains is hard or impossible to detect (e.g. in host-pathogen, antigen-antibody interactions) or where protein-protein interactions for the same polypeptide chains varies over cycles of mechanical activities. In this study, we theorise that a better integration between experimental data and predictions can be made possible by improving the template mechanism that is already present within AlphaFold.

We therefore present a simple, easy to use and effective tool to integrate experiments and predictions by modifying the mechanisms

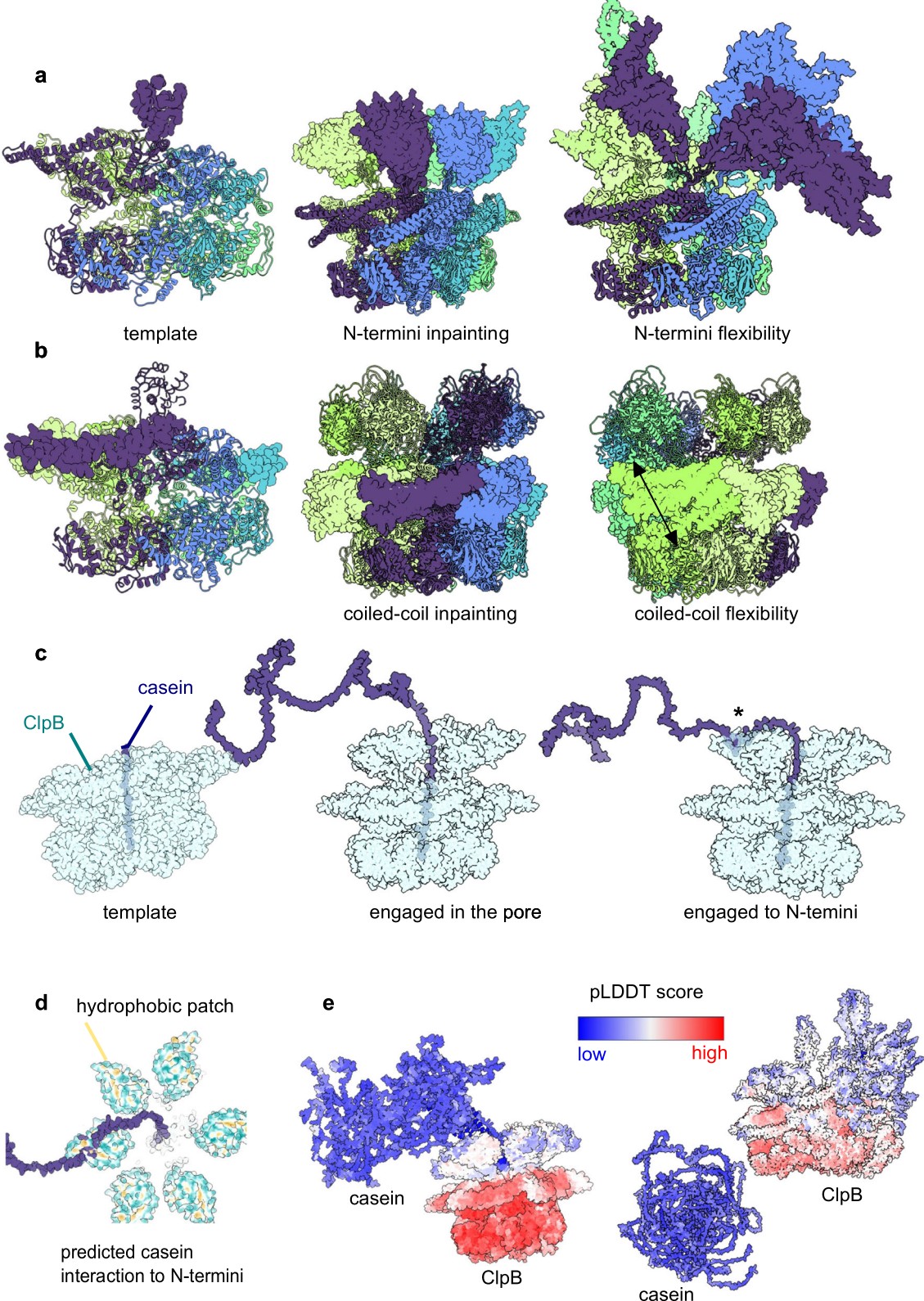

**Fig. 7 | Analysis of ClpB hexamer using AF_unmasked. a** Given template and inpainted N-termini. N-termini are shown as surfaces while other domains are as cartoon. Each subunit of the hexamer is coloured differently. **b** Inpainting on the M-domains, shown as surfaces. The arrow shows the possible motion of the M-domain. **c** Inpainting of the interaction between ClpB and casein. The asterisk shows the newly predicted interaction area. **d** View of the hydrophobic regions of ClpB termini interacting with casein. **e** Models of ClpB and casein and relative confidence scores.

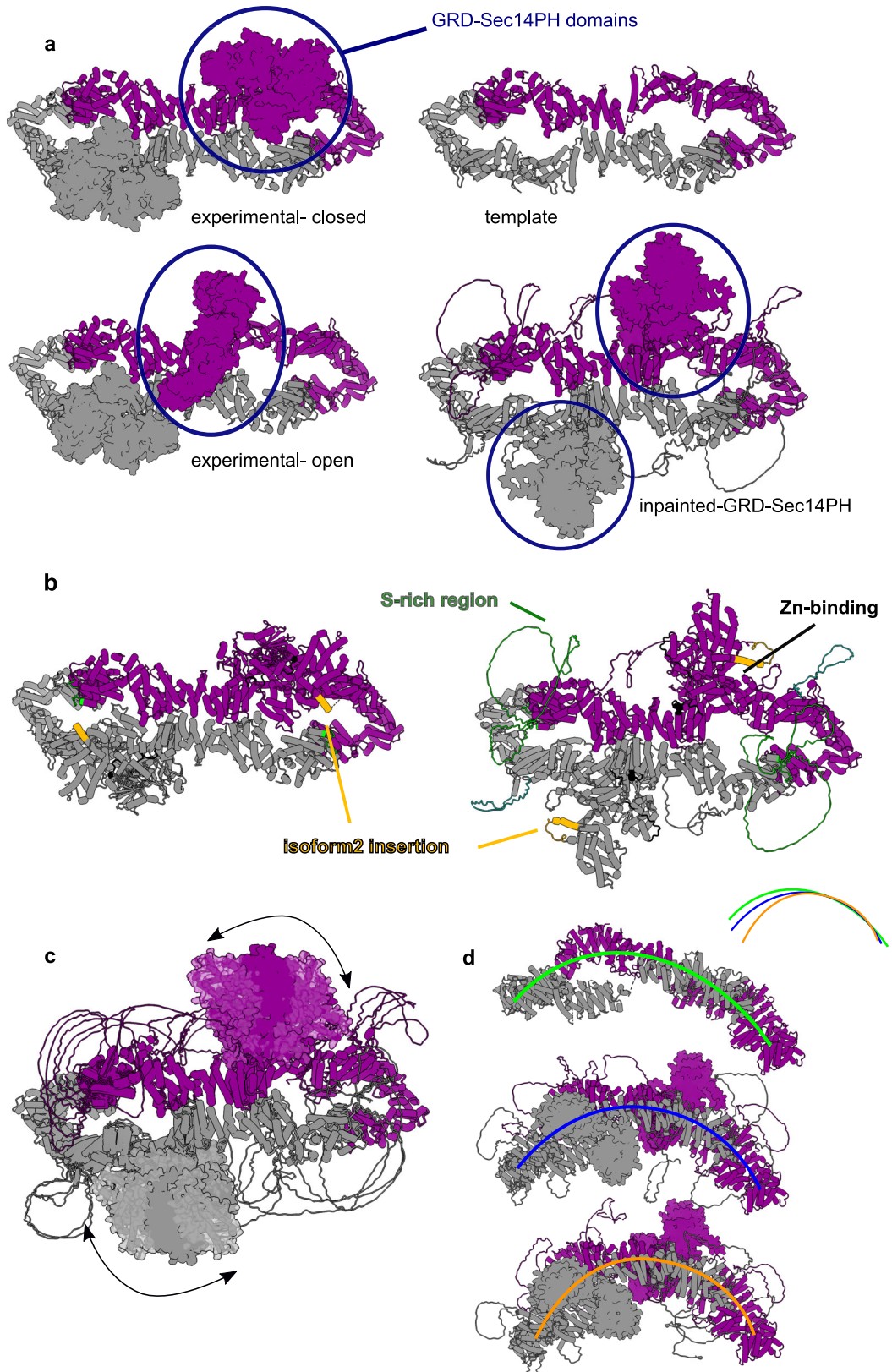

**Fig. 8 | Analysis of the Neurofibromin (NF1) dimer using AF_unmasked.**
**a** Inpainting of the GRD and Sec14-PH domain of NF1 in an intermediate conformation in between the experimentally observed closed and open states.
**b** Comparison of the closed experimental conformation of NF1 isoform 2 (on the left) with a AF_unmasked conformation. Important regions are highlighted.

**c** Superimposition of several AF_unmasked predictions where the GRD and Sec14-PH domain were modelled in intermediate positions, suggesting a possible motion path for these domains. **d** Three different AF_unmasked predictions showing a bending of the helical NF1 platform, also represented with differently coloured curved lines.

to embed structural templates within AlphaFold. We call this method AF_unmasked.

We tested AF_unmasked through a series of use cases and showed that it is effective not only when the templates are ideal (as in the PDB benchmark dataset), but even when they consist of homologs (Homologous PDB dataset), or rough models built from unbound monomers where amino acids at the interface are clashing (H1142), or when a mutation causes a conformational change (T1110o, T1109o). We could also perform *structural inpainting*, i.e. we let AlphaFold use evolutionary information from the MSA to fill in gaps in experimental structures (H1111, Rubisco, ClpB and Neurofibromin).

In all the scenarios we investigate, AlphaFold's own predicted quality score remains the best tool to evaluate the quality of the predictions. Even in cases where all evolutionary information is deleted and ideal templates are provided, the ranking confidence score is not confounded by the templates. Instead, the confidence score increases only in cases where the template puts AlphaFold closer to the solution, producing better predictions (Fig. S2). In other words, AlphaFold does not blindly trust the templates but rather, it is able to recognise good multimeric templates even in cases where it would not be able to produce good models on its own. In most cases, we expect that performing additional sampling will yield results that are even closer to the ground truth, and it will be relatively easy to select those results just by looking at the predicted quality metrics.

When inpainting large domains, we showed that AF_unmasked can sample the space of conformations more efficiently and predict them in multiple conformations. These conformations are very interesting for the interpretation of low-to-medium resolution cryo-EM and cryo electron tomography (cryo-ET) structures as well as for very flexible domains. We envision that intermediates states picked up by AF_unmasked could be of use for deducing correct movement paths underlying large domains' motions. From a general structural biology perspective, AF_unmasked allows to augment the available structural information in a number of ways (e.g. loops building, domain conformational variability) that can work as baseline for the design of novel biochemical and biophysical experiments.

Another characteristic of AF_unmasked is that it finally makes it possible to predict very large protein models in one go, since leveraging experimental data enables a reduction in the number of sequences in the MSA, thereby drastically reducing GPU memory and computational time requirements. This enables accurate protein structure predictions of very large protein complexes (up to 27 chains and 8460 amino acids in our experiments) without the need for external tools to perform docking of single subunits.

We believe that AF_unmasked, which is open source and easy to install and use, will be a valuable addition to the toolbox of all structural biologists.

## Methods

The changes to the standard AlphaFold pipeline are depicted in Fig. 1. In summary, a new template strategy is implemented where multimeric structures, either from released structures in the PDB or produced by the user, can be input at prediction time so that the positioning of protein chains with respect to each other informs the prediction task. Novel components are outlined more in detail below.

### The current implementation of templates in AlphaFold

The data pipeline in AlphaFold can select up to four structural templates. When predicting monomers, a template input is a set of coordinates of shape $L \times 37 \times 3$, where $L$ is the number of residues in the input sequence, 37 is the number of unique possible atom types for any type of amino acid, and 3 are the $x, y, z$ dimensions.

When predicting multimeric targets, AlphaFold's data pipeline will perform a separate template search for each unique input sequence. The first template is selected from the search results for each chain,

and coordinates are extracted from the corresponding PDB files. The coordinates are then mapped to a target sequence through a series of sequence alignments and then merged across all chains so that the template input to AlphaFold contains $x, y, z$ coordinates for all atoms covered by templates across all chains (Fig. 1, Template masking block). The coordinates are merged in a single list irrespective of whether the PDB hits originate from the same or completely different PDB structures. This step is performed up to four times, which is the maximum number of templates used in AlphaFold.

For all template inputs, AlphaFold's template-embedding module computes distogram features from the coordinate inputs. Distograms contain information about distances between all pairs of $C\beta$ atoms in a template. This information might describe distances between residues that sit in the same chain (intra-chain distances) as well as distances between residues that sit in different chains (cross-chain distances). At this stage, a *multichain mask* is applied to all distograms.

This is a boolean 2D array of shape $L \times L$ used to mask out sections of the distogram that contain cross-chain distances (Fig. 1). This masking is necessary since cross-chain distances obtained by merging chains from different PDB entries with no shared frame of reference are incorrect, so they should be ignored by the neural network.

### Using restraints from multi-chain templates

We introduce the possibility of either turning off multi-chain masking completely (Fig. 1, Unmasked label) or to switch to a complementary masking approach, where only cross-chain restraints are enforced in the template input while the intra-chain restraints are ignored. We accompany this modified neural network model with a set of tools to format and prepare ad-hoc multimeric templates. The templates can be either deposited PDB structures, as is done in classical homology modelling, but could also be drafts of experimental structures, for example when using AlphaFold as a tool to aid molecular replacement or for building structures from cryo-EM densities with different local resolution of details.

### Preparing ad-hoc templates

The template search in AlphaFold's standard pipeline is similar to other homology modelling tools, where a database of sequences in the PDB is queried with the target sequence or with a protein profile built from a target MSA. Whenever a template hit is found, the sequence alignment against the target sequence is used in the template pipeline in order to map atom coordinates from the PDB hit onto the target sequence. Gap regions in these alignments, i.e. sections of amino acids that are not covered by a given template, are masked so that they can be ignored by the neural network model at inference time.

The template preparation step in our pipeline replaces the default in AlphaFold by allowing the user to align a set of target sequences, or previously built monomeric structures, against a set of user-selected template structures. The target sequences that AlphaFold will model are mapped to the templates through an alignment step.

The alignment step can be done by sequence or by structure. In cases where target and template sequences are close homologs a sequence alignment might be enough, but when using remote homologs a structural alignment will be more accurate. Structural alignments can be performed by rigidly superimposing models of monomer chains onto the template with TM-align[47]. Another option is to use a superposition-free tool like lDDT-align[48], which was developed to perform structural alignments while maximizing the lDDT score. lDDT-align performs alignments of distance matrices rather than rigid-body superpositions, and therefore should produce better alignments between remote homologs or structures that undergo significant domain arrangements when binding to a partner. By default, sequence or structural alignments are written to the template alignment file that AlphaFold uses to correctly parse the templates (*pdb_hits.sto*), but may

also be used to roto-translate structures of each unit onto a template and merge the poses obtained this way into a new coarse template.

The templates do not need to cover the entire set of target structures in the complex, as AlphaFold will automatically mask out any target that is not included in the alignment. The user may even take advantage of multiple partial templates covering different parts of a target complex by using the structural superposition tools to generate a composite coarse template.

If a template is given in PDB format rather than mmCIF, it is automatically converted to mmCIF for compatibility with AlphaFold's structure parsing step. The preparation step can be performed up to four times, either with four different templates or by repeating the same template, to fill out all of the input template slots. We have observed that using all the template slots strengthens the impact of templates.

## Clipping of Multiple Sequence Alignments

The prediction of a structure is a balancing act between restraints derived from templates and those derived from MSAs. When the two are in disagreement with each other, and if the MSA-derived restraints are too strong, the Evoformer might ignore the template information, and vice-versa. Here, we introduce an option to manually reduce the number of sequences that should be included from each MSA to increase the influence of templates on the results. In this study, we test this approach to the extreme in our benchmarks (see Results) where we completely delete all evolutionary inputs and include the target sequence alone in the input MSA.

Clipping the MSA comes with the side benefit of greatly reducing the computational resources needed to generate each structure, as memory cost of the Evoformer scales with the squared number of sequences in the merged MSA[1]. This makes it possible to generate much larger structures than otherwise possible with the standard version of AlphaFold. When working with larger structures, we noticed that a smaller memory footprint considerably decreases the time needed to generate predictions. In this study, whenever we wish to generate structures for large multimers, we make sure we include only up to 512 sequences in the merged MSA input (Supplementary Table 1). Including fewer than 512 sequences will not further decrease the memory footprint during the prediction step, as smaller MSAs are zero-padded to a minimum depth of 512, and the evolutionary information is still useful to fill in information that might missing in the templates.

In all AF_unmasked tests, we always disable pairing of MSAs for different units in a complex, so that the cross-chain constraints are only derived from the templates rather than from evolutionary inputs. In hetero oligomers, this is done by clipping MSAs generated from the Uniprot database to a single sequence. In homo oligomers, we introduce a special flag (`–separate_homomer_msas`) to disable automatic pairing of MSAs from identical sequences.

## Inpainting of experimental structures

Due to flexibility or to incomplete or noisy data, many deposited protein structures have missing stretches of amino acids, in some cases even up to entire domains. We propose to use AlphaFold to predict missing regions by inputting partial experimental structures as templates, along with the full target sequence, and let AlphaFold sample the missing parts while using what is available in the template to restrain the search. While this is theoretically already possible with monomeric structures in AlphaFold, we are not aware of other studies attempting this, let alone with multimers.

In image processing, *inpainting* is the task of reconstructing missing parts of images, for example to reconstruct damaged parts of old photos or to remove watermarks. In the past few years, a number of deep learning tools have been developed for the task of inpainting protein structures. The first methods using inpainting in protein

structure prediction treat it as a classical image inpainting task, where missing areas in 2D protein distance maps are filled in using Generative Adversarial Neural Networks[49,50]. The idea was later generalised to directly filling in gaps in 3D structures in $RF_{joint}$[51], a modified version of RoseTTAFold[52] were masking fragments of protein sequence/structure pairs allows to predict native-like structures while generating novel sequences in the masked regions. In $RF_{joint}$, inpainting is intended to aid protein design (i.e. the main objective is to generate a novel sequence) and is mostly showcased on smaller monomers. A second 3D inpainting approach has been proposed to design complementarity-determining regions (CDRs) in antibodies[53]. Other approaches have shown that inpainting by diffusion models may be used to fill in small loops in known structures and suggest that the predictions may capture some conformational variability of interfacial residues in inpainted areas[54].

In this study, we extend structural inpainting of proteins to a larger scale, i.e. from fragments of monomers to large, possibly mobile domains in multimers. In contrast to previous methods, we show that it is not necessary to train a specialised neural network to achieve this. Rather, we use templates to restrain prediction around a desired prior while letting AlphaFold's learned potential to sample areas of uncertainty. In this work, we inpaint structures in a number of use cases, i.e. to complete existing experimental structures with missing regions, but also to predict the impact of single-point mutations and to gain insights into dynamics of whole domains in large multimers.

## PDB dataset

In order to assess the efficacy of the new template strategy, we use Dockground[55] to obtain a list of 633 heterodimers in the PDB that were released after December 2022, when the latest version of AlphaFold multimer parameters (v2.3) was made public. We perform redundancy reduction of these structures by comparing all-against-all with MM-align[56] and keeping only one representative when their TM-score is above 0.4. The redundancy-reduced set is made of 251 structures.

## Homologous PDB dataset

In order to assess whether AF_unmasked is able to transfer cross-chain template information between homologous complexes, we also produce a set of homologous templates starting from the aforementioned PDB dataset.

We select from the initial PDB dataset a number of targets where AlphaFold fails to produce a good, top-ranked prediction (DockQ below 0.2 for prediction with highest ranking confidence score). Then, we use Foldseek[57] to find protein complexes in the PDB where two chains are homologous to the targets and use DockQ to select templates against the native conformation (DockQ between template and target above 0.15). The selection step with DockQ is necessary to make sure that the templates are not completely incorrect to cut down on computational costs.

In this way, 28 of the 251 targets in the initial PDB dataset are selected, along with the lists of homologous templates. These represent structures where the default version of AlphaFold could not produce an acceptable top-ranked prediction while useful templates could be found in the PDB.

## CASP15 examples

We select three examples of challenging multimeric targets from CASP15 to showcase different features of AF_unmasked.

For all these targets, we use the same MSA inputs that were generated by the baseline server NBIS-AF2-multimer[5].

We compare predictions made with AF_unmasked for these targets against those made by the "Wallner" group in CASP15, a top-performing predictor in the multimer category[58]. This allows to assess the effect of the template strategy introduced with AF_unmasked while controlling for other variables, such as neural network models

(AlphaFold-Multimer, v2.2), dropout sampling strategy and MSA inputs, since Wallner also used those generated by NBIS-AF2-multimer[58].

It is important to note here that the aim of these tests is not to prove that AF_unmasked performs better than other CASP15 methods, but rather, to demonstrate the effectiveness of our approach in accurately predicting difficult or large targets when the right template is used. For example, we will be using information that was not accessible to the predictors at CASP15, such as partial or full native structures of targets that have been released since.

### Analysis of cryo-EM macromolecular assemblies

We reason that AF_unmasked will be useful in the analysis of large macromolecular complexes obtained by cryo-EM. While AlphaFold in the current form, both in the monomer and in the multimer implementations, is able to answer structural questions about small and stable protein in a satisfactory way, it does not address the structural complexity of large and flexible macromolecules, a typical type of cryo-EM target. This method will therefore aid in the modelling of (i) poorly resolved parts of cryo-EM density maps, either because of flexibility or low occupancy, and (ii) large asymmetrical homo-oligomers which display variability in domain positioning. To test these scenarios, we consider three different large complexes: the Rubisco enzyme, the ClpB disaggregase and the Ras-regulator Neurofibromin. All these targets are large homo-oligomers that cannot be correctly predicted with the standard version of AlphaFold-Multimer. They also have flexible domains that are poorly or not resolved in the experimental cryo-EM densities.

### Sampling strategy and other settings

We use the AlphaFold code as released in version 2.3. In the initial tests done with the PDB sets, we use the latest release of Alphafold-Multimer parameters (v2.3) so that we could make a fair comparison against the latest version of AlphaFold-Multimer. We also used v2.3. parameters when modelling larger complexes, as we expected them to speed up inference times (targets: H1111, neurofibromin)[20]. When comparing against other predictors at CASP15 we use the parameters released in v2.2, as v2.3 parameters had not been released then, so that we could eliminate the variable of using different neural network models when making a direct comparison. In general, v2.2 parameters seem to incorporate well the templates in most cases, so we used those by default on all other tests. While in most cases we only generate 25 predictions (5 per neural network model), in some cases where inpainting is performed and we wish to assess its impact on the predictions (targets: H1142, T1109/T1110, neurofibromin) we choose to stimulate additional sampling of a target's conformational space by turning on dropout at inference time following a strategy similar to the Wallner group's in CASP15[58] and predicting up to 200 structures per neural network model (1000 in total). We also increase the number of recycles from the default for v2.2 (3) to 20, as in v2.3. Finally, we introduce an option that forces AlphaFold to use separate MSAs and template for all chains, even those with identical sequences, so that the correct templates can be used in the case of homomers. Relaxation is turned off for all experiments to isolate contributions of the neural network from those made by the relaxation protocol using the AMBER force field.

### Scoring

We rank predicted models by AlphaFold's default ranking metric, i.e. the ranking confidence, which for multimers is the weighted sum of predicted TM-score ($pTM$) and interface pTM ($ipTM$) with weights of 0.2 and 0.8, respectively. In cases where we want to assess the predicted quality of a specific area of a structure, we average AlphaFold's predicted Local Distance Difference Test (pLDDT) for the amino acids in that area. When comparing a prediction to a native structure in order to assess its quality, we use the DockQ score[19] (v2.0), a measure of the quality of the predicted interfaces in a complex as a continuous measure ranging from zero to one. To describe a structure prediction qualitatively, we also follow the DockQ quality classification, according to which a prediction can be: Incorrect (DockQ score below 0.23), Acceptable (DockQ between 0.23 and 0.49), Medium (DockQ between 0.49 and 0.8), High quality (DockQ score above 0.8)[19]. In cases where we want to assess the quality of the conformation at the interface alone, we use the interfacial residue RMSD (iRMSD) as calculated by DockQ. In cases where we calculate the RMSD over all atoms of two superimposed molecules, we use TM-score[59].

Q-scores, as described by Pintilie et al.[60], were employed to evaluate the fit between proposed protein structures and their corresponding cryo-EM density maps. Q-scores quantify the resolvability of the cryo-EM map based on the fit of the model within the density. When comparing multiple structural models within the same density, a higher Q-score signifies a superior fit, indicating that the model enables the construction of higher resolution models from the map data. We use the QScore plugin (v1.1) for UCFS ChimeraX (v1.71) to derive Q-scores from predictions and Cryo-EM densities.

### Reporting summary

Further information on research design is available in the Nature Portfolio Reporting Summary linked to this article.

## Data availability

The source data underlying Figs. 2, 3c, 5c–d and Supplementary Figs. S1–S15 are provided as a Source Data file. All the predicted structures, along with log files, inputs and templates are available with [https://doi.org/10.17044/scilifelab.24198669]. ChimeraX sessions for the analyses of Cryo-EM test cases are available with [https://doi.org/10.17044/scilifelab.25653297]. Supplementary movies referenced in text are available as Supplementary data files. Deposited PDB structures referenced in the text are: 7ALW, 7QIJ, 4K2H, 5IU0, 1QVR, 5OG1, 6RN3, 7PGU, 6OB3, 1NF1, 3PEG. Source data are provided with this paper.

## Code availability

The source code, installation instructions, user manual and examples are available on GitHub: github.com/clami66/AF_unmasked, [https://doi.org/10.5281/zenodo.13364959][61]. A Jupyter notebook is available to run the tool on Google Colab or similar: [https://github.com/clami66/AF_unmasked/tree/notebook].

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

## Acknowledgements

This work was funded by SciLifeLab as a Technology Development Project (BeyondFold), and by the SciLifeLab platforms National Bioinformatics Infrastructure Sweden (NBIS), Cellular and Molecular Imaging (CMI), and Integrated Structural Biology (ISB). CM is financially supported by the Knut and Alice Wallenberg (KAW) Foundation as part of NBIS at SciLifeLab. SA is financially supported by KAW grant KAW2021.0347 to MC. MC is financially supported by SciLifeLab as part of the Cryo-EM National Infrastructure Unit and by the Stiftelsen för Strategisk Forskning (SSF) grant RIF-21. BW is financially supported by KAW as part of the WASP-DDLS joint program. Computations were performed at NSC Tetralith provided by the National Academic Infrastructure for Supercomputing in Sweden (NAISS) and the Swedish National Infrastructure for Computing (SNIC), partially funded by the Swedish Research Council through grant agreements no. 2022-06725 and no. 2018-05973, and at NSC BerzeLiUs provided by the National Supercomputer Centre (NSC) and funded by KAW. We would also like to thank Dr. Nicholas Pearce, Dr. Yogesh Kalakoti, Dr. Tim Schulte, Dr. Piotr Draczkowski and Nicholas Debouver for useful feedback throughout the study and during the development of manuscript and code.

## Author contributions

C.M. and B.W. conceived the study. C.M. designed and implemented the tool, performed analyses of results, developed the manuscript, supervised the project. B.N. supervised the project development and provided funding. S.A. and M.C. performed analyses of Cryo-EM test cases. M.C. supervised the Cryo-EM analyses and developed the manuscript.

## Funding

## Competing interests

The authors declare no competing interests.
