## [Peer Review File · Nature Communications]

Unmasking AlphaFold to integrate experiments and predictions in multimeric complexesEditorial Note: This manuscript has been previously reviewed at another journal that is not operating a transparent peer review scheme. This document only contains reviewer comments and rebuttal letters for versions considered at *Nature Communications*.

REVIEWER COMMENTS

Reviewer #1 (Remarks to the Author):

This revised version of AlphaFold Unmasked addresses most of my concerns from the previous version. Some remaining comments:

1, One new concern is the use of the term, "near-perfect" in the abstract.

The abstract states that, "This new approach generates near-perfect structures even when little to no evolutionary information is available and imperfect experimental structures are used as a starting point". It might be good to use some other words, as "near-perfect" makes it sound as though a result is nearly perfect, which would imply that it should be very nearly as useful as a perfect model, which is certainly not true in many circumstances.

Later in the paper "near-perfect" seems to be defined as, "a near-perfect (DockQ score>0.8) prediction". It would probably be best to define "near-perfect" or whatever term is used right in the abstract.

2. As pointed out in the previous review, the authors note that, "Lastly, we generate a third set of predictions by deleting clashing sets of residues from the templates and letting AlphaFold inpaint these areas. This seems the best approach resulting in even more diverse predict"

It would be helpful to know how users intended to carry out this process? It seems rather time-intensive and manual for automated docking.

Reviewer #1 (Remarks on code availability):

I didn't review the code again because last time the installation crashed.

Reviewer #3 (Remarks to the Author):

I have been tasked to judge the response to reviewer and revised version of the manuscript after a reviewer withdrawal. The authors broadly responded to the reviewer's concerns. I still see a couple of minor concerns and what is for me a major point missing in the discussion.

Major point:

the approach of inpainting with deep learning protein prediction is not fully novel, as the authors do acknowledge. The authors should really compare their approach with wang et al 2022 Science. I understand a systematic comparison of the casp15 set, which is desirable, may no longer be possible or fair at this stage. At least I would suggest editing the discussion to compare them and discuss what is novel about the authors' approach. I understand that wang et al 2022 Science is cited, but a comparison beyond "rosettafold vs alphafold" would really benefit the paper. Even if it is just a small section of the discussion at this point.

Another preprint from 2022 with inpainting of CDRs

<https://www.biorxiv.org/content/10.1101/2022.07.09.499440v1.full> and a specific module for inpainting <https://www.biorxiv.org/content/10.1101/2023.11.21.568057v1.full.pdf> should probably also be cited, if not discussed. In short, I do not believe that it is these authors that "introduce structural inpainting" as they claim at the end of the introduction or hint at in the abstract. This does not reduce their technical accomplishment but it is an exaggerated claim in my view.

Minor concerns

rubisco: the authors have not addressed the concern of reviewer 2 for quantifying the fit to experiment: cross-correlation over the particular region should be reported, not a visual fit that is very hard to judge on a transparent 2d figure.

The authors did not respond to the reviewer question on MSA content in the sense that they did not calculate the number of effective sequences (Neff as in this <https://academic.oup.com/bioinformatics/article/31/5/674/2748147> and the original alphafold paper) as an estimate of the richness of the msa in each case.

Reviewer #3 (Remarks on code availability):

The code is fully open source, documented and available. The README is well put together and makes the installation steps clear.

We would like to once more acknowledge and thank the reviewers for their comments.

Reviewer #1 (Remarks to the Author):

This revised version of AlphaFold Unmasked addresses most of my concerns from the previous version. Some remaining comments:

1, One new concern is the use of the term, “near-perfect” in the abstract.

The abstract states that, "This new approach generates near-perfect structures even when little to no evolutionary information is available and imperfect experimental structures are used as a starting point". It might be good to use some other words, as “near-perfect” makes it sound as though a result is nearly perfect, which would imply that it should be very nearly as useful as a perfect model, which is certainly not true in many circumstances.

Later in the paper “near-perfect” seems to be defined as, "a near-perfect (DockQ score>0.8) prediction". It would probably be best to define “near-perfect” or whatever term is used right in the abstract.

We agree with the reviewer that the terminology was unclear. We have added a definition of these qualitative descriptors in the methods. We have also changed the descriptors so that they follow those established in the DockQ paper (Incorrect, Acceptable, Medium and High quality).

2. As pointed out in the previous review, the authors note that," Lastly, we generate a third set of predictions by deleting clashing sets of residues from the templates and letting AlphaFold inpaint these areas. This seems the best approach resulting in even more diverse predict”

It would be helpful to know how users intended to carry out this process? It seems rather time-intensive and manual for automated docking.

We add to the text to explain how the inpainting process of clashes is automatic and only requires that the user enables this option by setting the appropriate flag during the template preparation step. The inpainting of clashes takes as much time as a regular prediction both during template preparation and inference steps.

Reviewer #1 (Remarks on code availability):

I didn't review the code again because last time the installation crashed.

We have added instructions on github on how to install AF_unmasked on machines with M1/M2 architecture. We believe that the reason for the crash was due to the fact that some dependencies are not available for this particular architecture. We tested this on a M2 machine, and it should be enough to launch the conda installation step in x86 compatibility mode:

```
“CONDA_SUBDIR=osx-64 conda env create --file=environment.yaml”
```

Reviewer #3 (Remarks to the Author):

I have been tasked to judge the response to reviewer and revised version of the manuscript after a reviewer withdrawal. The authors broadly responded to the reviewer's concerns. I still see a couple of minor concerns and what is for me a major point missing in the discussion.

We would like to thank reviewer #3 for taking on the review of our manuscript at this late stage. We would also like to apologise for not making a highlighted version of the manuscript available from the beginning. All changes from the previous version are highlighted on the new one.

Major point:

the approach of inpainting with deep learning protein prediction is not fully novel, as the authors do acknowledge. The authors should really compare their approach with wang et al 2022 Science. I understand a systematic comparison of the casp15 set, which is desirable, may no longer be possible or fair at this stage. At least I would suggest editing the discussion to compare them and discuss what is novel about the authors' approach. I understand that wang et al 2022 Science is cited, but a comparison beyond "rosettafold vs alphafold" would really benefit the paper. Even if it is just a small section of the discussion at this point.

Another preprint from 2022 with inpainting of CDRs

<https://www.biorxiv.org/content/10.1101/2022.07.09.499440v1.full> and a specific module for inpainting <https://www.biorxiv.org/content/10.1101/2023.11.21.568057v1.full.pdf> should probably also be cited, if not discussed. In short, I do not believe that it is these authors that "introduce structural inpainting" as they claim at the end of the introduction or hint at in the abstract. This does not reduce their technical accomplishment but it is an exaggerated claim in my view.

We agree with the reviewer on this point. Although we did not mean to suggest that we were the first to introduce structural inpainting, we can see that it was not clear enough in the text. We have rephrased the abstract and introduction to make this absolutely clear.

We have also expanded the methods section on inpainting, detailing how previous methods focus mostly on inpainting as a way to design new sequences, rather than completing available structures, and are usually limited to (or at least, they were showcased on) small stretches of amino acids. We thank the reviewer for pointing out the two preprints on inpainting approaches, which we were not aware of, and we have included them in the discussion. We believe the text has considerably improved as a result.

Minor concerns

rubisco: the authors have not addressed the concern of reviewer 2 for quantifying the fit to experiment: cross-correlation over the particular region should be reported, not a visual fit that is very hard to judge on a transparent 2d figure.

We have added a quantitative measure of the fit between predictions and the map. We use a cross-correlation based measure (Q-score) to compare the fit between the best AF_unmasked and standard AF-Multimer predictions and the map. We opt for Q-scores rather than pure cross-correlation, as in our experience the latter performs poorly at discriminating between models in low-density regions of the map.

The authors did not respond to the reviewer question on MSA content in the sense that they did not calculate the number of effective sequences (Neff as in this <https://academic.oup.com/bioinformatics/article/31/5/674/2748147> and the original alphafold paper) as an estimate of the richness of the msa in each case.

We have added a table in the supplementary detailing the number of hits that were used from each database search (Uniref90, BFD/Uniref30, Mgnify, and Uniprot) for each target. While we agree that Neff might be a better measure of the amount of information allowed to flow from the MSA to the neural network, it is not straightforward to calculate it in the case of multimers, as the final MSA is a block composition of multiple MSAs (Nx4 MSAs, where N is the number of units in the multimer) calculated at runtime by AlphaFold. We believe that the parameters as we detail them in our table nonetheless will help reproduce our results. We also clarify the strategy that we used to select these parameters. Binary representations of all MSAs used in the experiments are available in the pickle files referenced in the data deposition section of the manuscript (DOI: 10.17044/scilifelab.24198669).

Reviewer #3 (Remarks on code availability):

The code is fully open source, documented and available. The README is well put together and makes the installation steps clear.

REVIEWERS' COMMENTS

Reviewer #1 (Remarks to the Author):

The authors have addressed all my concerns.

Reviewer #3 (Remarks to the Author):

The authors have addressed my concerns and have presented a much improved text. I recommend the paper be accepted.